# Mitigating Social Biases in Text-to-Image Diffusion Models via Linguistic-Aligned Attention Guidance

## ABSTRACT

Recent advancements in text-to-image generative models have showcased remarkable capabilities across various tasks. However, these powerful models have revealed the inherent risks of social biases related to gender, race, and their intersections. Such biases can propagate distorted real-world perspectives and spread unforeseen prejudice and discrimination. Current debiasing methods are primarily designed for scenarios with a single individual in the image and exhibit homogenous race or gender when multiple individuals are involved, harming the diversity of social groups within the image. To address this problem, we consider the semantic consistency between text prompts and generated images in text-to-image diffusion models to identify how biases are generated. We propose a novel method to locate where the biases are based on different tokens and then mitigate them for each individual. Specifically, we introduce a Linguistic-aligned Attention Guidance module consisting of Block Voting and Linguistic Alignment, to effectively locate the semantic regions related to biases. Additionally, we employ Fair Inference in these regions to generate fair attributes across arbitrary distributions while preserving the original structural and semantic information. Extensive experiments and analyses demonstrate our method outperforms existing methods for debiasing with multiple individuals across various scenarios.

## CCS CONCEPTS

• **Applied computing**; • **Computing methodologies → Computer vision**;

## KEYWORDS

Text-to-image generation, Social biases, Diffusion model.

## 1 INTRODUCTION

Text-to-image generative methods develop rapidly based on the diffusion models [19, 35, 42, 48], achieving impressive generation fidelity and diversity on a large range of tasks including image editing [5, 9, 16, 18, 32], style-transfer [12, 53, 59], concept learning [24, 26, 41, 56] and so on. However, these powerful models have revealed severe potential risks of social biases such as gender, race, and their intersections when conditioned on text prompts describing human-related content including occupations, personality traits, or simply the term "person" [3, 8, 31, 31, 45]. As shown

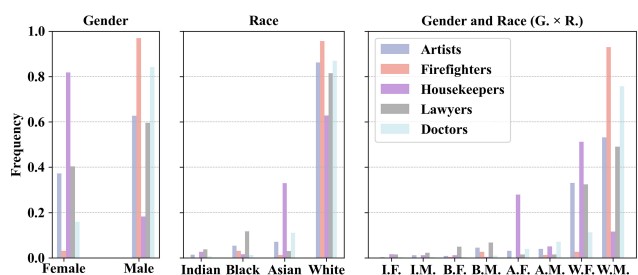

**Figure 1: The frequency of different social groups across various occupations generated by Stable Diffusion, where I., B., A., and W. represent Indian, Black, Asian, and White respectively, and F. and M. represent Female and Male respectively.**

in Fig. 1, the classic text-to-image generative model Stable Diffusion [39] demonstrates biases toward "Male" and "White" in most occupations, whereas, for housekeeper, the model skews toward "Female". A similar pattern is observed in DALLE-v2 [8, 31]. These biases may propagate a distorted worldview and spread unforeseen prejudice and discrimination against certain groups.

Some progress has been made to mitigate these biases for diffusion-based text-to-image generative models. Most of them [10, 15, 22, 25, 25, 32, 46] re-train or fine-tune different components of the original model, necessitating expensive computational resources. Other works [2, 11] are training-free and only require fair prompts for debiasing. However, all these debiasing methods are designed for scenarios involving only one individual. When applied to more general contexts where multiple individuals appear in one image, they often produce homogeneous social groups. As shown in Fig. 2, current debiasing methods exhibit almost the same race or gender to all individuals in an image, which harms the diversity of social groups within the image. In contrast, our method enables individualized debiasing and exhibits diverse genders and races.

To address the challenge of debiasing for more general scenarios with multiple individuals and representing diverse social groups within the image, we consider how these biases are generated. For text-to-image generation, the generated results are semantically instructed by the conditioned text prompts, benefitting from the architecture of the diffusion model [39]. This property indicates that the tokens of the prompt are accountable for the semantic information within the image [23, 49], potentially introducing biases. For instance, tokens like "faces" and "doctors" may be associated with certain social groups in the generated results, such as "Male" and "White", which introduce biases for each individual related to them. By identifying the semantic regions associated with these tokens, we can locate where the biases are and subsequently debiasing them for each individual.

In this work, we introduce a Linguistic-aligned Attention-Guided Fair Generation Model, which locates the semantic regions related

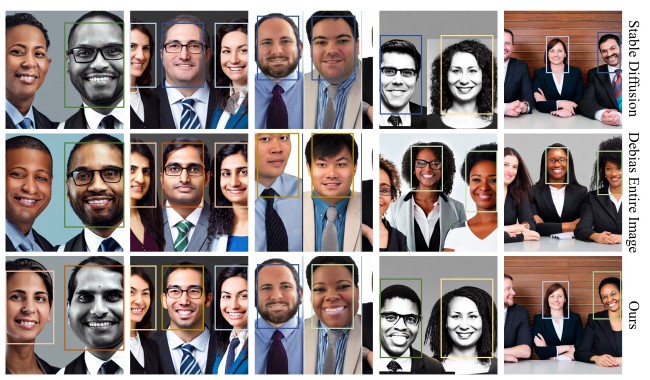

**Figure 2: Results of generated "lawyers" of different methods. Current methods debias across the entire image, exhibiting homogenized attributes within the image, while our method handles multiple individuals in the same image with diverse genders and races.**

to biases based on different tokens and generates fair attributes in these regions to mitigate biases. To enhance the accuracy and robustness of the extracted semantic regions, we introduce a Linguistic-aligned Attention Guidance module consisting of Block Voting and Linguistic Alignment. Block Voting aims to obtain more region-effective cross-attention maps and avoid disturbance of global information. Linguistic Alignment facilitates robust and stable semantic regions from different thresholds by maintaining the linguistic correlations of different tokens. Once the semantic regions are obtained, we further employ Fair Inference to ensure each individual exhibits fair attributes while preserving the original structural and semantic information. Furthermore, the debiasing for each individual is achieved by confirming the target distributions of social groups. Given intended fair prompts, such as gender and race, our method achieves fair generation in more general scenarios including multiple individuals within the image and is free of training.

Our contribution can be summarized as follows:

- We introduce a Linguistic-aligned Attention-guided Fair Generation Model to mitigate social biases in general contexts including multi-face scenarios. To the best of our knowledge, this is the first debiasing method capable of dealing with multiple individuals within the image.
- We propose a Linguistic-aligned Attention Guidance module to effectively locate the semantic regions containing biases and Fair Inference to achieve fair attributes for each individual while preserving original structural and semantic information.
- Comprehensive analysis and experiments are conducted to demonstrate the superiority of our method in mitigating biases with multiple individuals across various scenarios.

## 2 RELATED WORK

### 2.1 Text-to-image Generation.

Text-to-image generative methods [13, 34, 54] used to based on Generative Adversarial Networks (GANs) [17]. However, these methods would encounter problems such as unstable training and suboptimal generation quality. With the development of diffusion models [19, 48], generative results with greater fidelity, higher resolution, and more diverse conditions under stable training have been achieved. Stable Diffusion [39], trained on large-scale text-image dataset [44], exhibits impressive performance on text-to-image generation, super-resolution, and unconditional image generation. Furthermore, various text-to-image generation tasks such as image-editing [1, 5, 16, 18, 30, 32], style-transfer [12, 59] and concepts learning [24, 41] are also achieved and exhibit impressive performance based on diffusion architecture.

### 2.2 Biases in Diffusion Models.

Pretrained on large image-text dataset [44] selected from the Internet without adequate filtering and labeling, many text-to-image diffusion models [37, 39] are discovered to present potential harmful biases on social dimensions such as gender, race, culture, and so on. For generating prompts describing occupations, personality traits, or simply the word "person", Stable Diffusion [39] and DALLE-v2 [37] present severe biases excluding groups of people [31]. Luccioni *et al.* [29] and Friedrich *et al.* [11] also uncover the gender bias for different occupations in diffusion models. Similarly, Bianchi *et al.* [3] discover that Stable Diffusion exhibits racial, ethnic, and gendered stereotypes and stereotype amplification on neutral prompts. Seshadir *et al.* [45] present that Stable Diffusion would amplify biases from training data between gender and occupations. Furthermore, Cho *et al.* [8] discover that Stable Diffusion tends to generate males and skin tone centered on few tones for occupations. Liu *et al.* [28] uncover the cultural stereotypes with negative impacts on various groups.

### 2.3 Bias Mitigation for Diffusion Models.

Existing bias mitigation methods focus on gender, race, and their intersections for occupations. Most of these methods finetuning certain components of the original model [10, 15, 22, 25, 25, 32, 46]. Shen *et al.* [46] employ biased direct fine-tuning for the sampling process with a distributional alignment loss. Esposito *et al.* [10] fine-tune text-to-image models on a synthetic fair dataset. Li *et al.* [25] trained a semantic h-space to achieve fair generation. Kim *et al.* [22] employ attribute classifiers to prompt-tune the text-to-image models for debiasing, and Li *et al.* [25] train a mapping network for text embeddings to guide the fair generation. TIME [32] and UCE [15] update the cross-attention layers to achieve semantic-aligned text-to-image editing and hence mitigate social biases. Other works [2, 11] utilize fair prompts for debiasing, which are free of re-training or fine-tuning the original model. Bansal *et al.* [2] incorporate ethical prompt intervention to original prompts. Fair-diffusion [11] employs semantic dimensions [4] to guide debiased generation with a given fair prompt table. However, the above methods focus on scenarios involving only one individual within the images, which harms the diversity of social groups for general contexts with multiple individuals.

## 3 METHOD

To achieve debiasing for each individual, our method consists of two steps, as illustrated in Fig. 3. We first obtain the semantic regions from text prompt tokens via the Linguistic-aligned Attention

**Figure 3: The pipeline of our method. In step one, we employ Linguistic-aligned Attention Guidance to obtain more accurate semantic regions relative to biases. In step two, we employ Fair Inference to generate fair attributes in these regions while persevering original structural and semantic information.**

Guidance module. After that, we employ the Fair Inference to debias each individual while preserving the original semantic and structural information. Note that our method can achieve arbitrary target distributions without re-training or fine-tuning the original model.

## 3.1 Preliminary

**Classifier-free Guidance.** Text-to-image generative models based on Stable Diffusion [39] employ the classifier-free guidance [20] to generate images conditioned on text prompts without additional pre-trained classifiers. During training, the text prompts $c_p$ drops randomly to facilitate unconditional and conditional objectives. During inference, the conditioned text prompts are sent to the noise predictor $\epsilon_\theta$ along with the latent code $z_t$ to predict noise at different timestep $t$ as follows:

$$\widetilde{\epsilon}(z_t, c_p) = \epsilon_\theta(z_t) + s_g(\epsilon_\theta(z_t, c_p) - \epsilon_\theta(z_t)), \quad (1)$$

where $s_g$ is the guidance scale determining the extent of text-guided instructions. The backbone $\epsilon_\theta$ is a conditional UNet [40] to predict the added noises.

**Attention in Diffusion Models.** The attention maps in self- and cross-attention layers of the diffusion UNet are discovered to characterize semantic information such as spatial layout, segmentations, and objects' shapes [16, 33, 38, 49], which stem from the fully convolutional nature of the UNet [14]. The self-attention maps model the correlations between different patches of the latent code, reflected in the generated image. While the cross-attention

maps present the correlations between each text prompt token and generated image patches. Therefore, these attention maps can be employed to obtain the semantic segmentations of the generated images and assess the relevance of image patches to specific words [16, 49].

## 3.2 Linguistic-Aligned Attention Guidance

To understand how biases are generated and locate them, we analyze the text prompt and concentrate on the different tokens. As for the text prompt "A photo of the faces of {number} {occupations}", the nouns "faces" and "{occupations}" are semantically responsible for biases, as the text-to-image model predominately associates these tokens with certain social groups, such as "Male" and "White", which introduces biases for each individual related to them.

**Semantic Regions.** Our goal is to locate the semantic regions associated with these tokens. To achieve this, we first obtain global semantic segmentation maps from self-attention maps which contain rich information about spatial layouts and object shapes [16, 33]. The self-attention maps act as similarity maps, correlating different patches of generated images and generating semantic clusters [16] through spectral clustering [47, 51] as $SC^c = F(SA, c)$, where $SA$ is the self-attention maps of a certain resolution and $F$ is spectral clustering. $SC^c$ are the semantic clusters with the number of $c$ clusters presenting different semantic segmentations.

To obtain the semantic regions $SR$ for different tokens $t$, cross-attention maps are employed as similarity maps denoting correlations between tokens and generated image patches to correlate different tokens and semantic clusters as follows:

$$SR_t = \sum_i^c (SC^i) \mathbb{1}_\tau \left( \frac{\sum(SC^i \cdot CA_t)}{\sum(SC^i)} > \tau \right), \qquad (2)$$

where $CA_t$ is the cross-attention map of token $t$. $\frac{\sum(SC^i \cdot CA_t)}{\sum(SC^i)}$ is the agreement score [33]. The $SC^i$ is labeled as part of the semantic region for token $t$ if the agreement score is larger than threshold $\tau$.

However, recent work discovers that averaged cross-attention maps can exhibit failures such as undesired attention to the corresponding area, denoted as attention leakage [55]. We observe similar failures in our setting, as shown in Fig. 4 (a), the original cross-attention maps of the token "faces" almost covered the whole image, which may lead to the inaccurate assignment of semantic regions. We further assess the effectiveness of cross-attention maps for semantic regions from different cross-attention blocks and timesteps. The metric denotes facial region accuracy evaluated by IoU scores between semantic regions of different cross-attention maps and facial regions detected by MTCNN face detector [57]. As shown in Fig. 4 (b), there are blocks more effective for semantic regions in early steps while some almost fail to work. This may stem from some attention blocks being more sensitive to objects' shapes while others mainly contain global information like textures, with limited contribution to local semantics. Therefore, averaging all cross-attention maps may result in undesired attention assignments. Besides, obtaining accurate semantic regions may require different threshold $\tau$ following Eq. 2. As shown in Fig. 4 (c), we evaluate the maximum facial region accuracy for the semantic regions of the nouns "faces", "{occupations}", denoted as biased tokens $B_t$ and the modifiers "the", "of" and "{number}", denoted as modifier tokens $M_t$, which are linguistically relative to $B_t$. The facial region accuracy corresponding to different tokens varies depending on different thresholds, indicating less robust and stable acquisition of these regions.

Based on these observations, we aim to obtain more accurate semantic regions and mitigate the above failures. We propose Block Voting to obtain cross-attention maps aligned with the semantic regions from region-effective blocks. Linguistic Alignment is employed to achieve a more robust and stable acquisition of semantic regions by linguistically correlating different tokens.

**Block Voting.** To mitigate attention leakage, we evaluate whether the cross-attention maps from different blocks and steps effectively represent semantic regions. The cross-attention maps are checked as follows to minimize disruptions of global information.

$$BV(CA_t^{b,s}) = \begin{cases} reject, & if \sum_i^c (SC^i) \mathbb{1}_\tau \left( \frac{\sum(SC^i \cdot CA_t^{b,s})}{\sum(SC^i)} > \tau_a \right) \in \{SC^c, 0\}. \\ CA_t^{b,s}, & otherwise. \end{cases} \qquad (3)$$

where $t$ is token, $b$ is attention block and $s$ is steps. If all semantic clusters or no clusters are chosen, indicating $CA_t^{b,s}$ hardly obtain effective semantic regions, the $CA_t^{b,s}$ will be rejected. After block voting, accepted $CA_t^{b,s}$ are summarized across different steps and

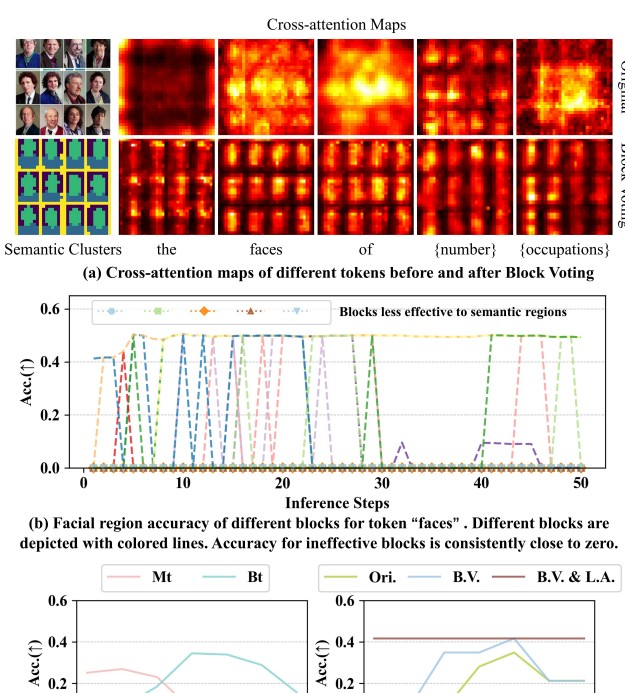

Cross-attention Maps

Semantic Clusters    the    faces    of    {number}    {occupations}

**(a) Cross-attention maps of different tokens before and after Block Voting**

**(b) Facial region accuracy of different blocks for token "faces". Different blocks are depicted with colored lines. Accuracy for ineffective blocks is consistently close to zero.**

**(c) Facial region accuracy of different tokens**    **(d) Effects of the proposed modules**

**Figure 4: (a) Variations in cross-attention maps before and after Block Voting. (b) The effectiveness of cross-attention maps across different blocks and steps. (c) The influence of different thresholds on the effectiveness of semantic regions corresponding to different tokens. (d) The proposed modules enhance the robustness and stability of obtaining semantic regions.**

scaled to the same resolution of different blocks, and then averaged.

$$\tilde{CA}_t = \frac{1}{B} \sum_b Scale_r \left( \sum_s BV(CA_t^{b,s}) \right). \qquad (4)$$

The final semantic regions $SR_t$ can be obtained with more region-aligned cross-attention maps $\tilde{CA}_t$ following Eq. 2. As shown in Fig. 4 (a), after Block Voting, the cross-attention maps are more consistent with the semantic clusters and identical to different semantic regions.

**Linguistic Alignment.** To generally obtain effective semantic regions, we aim to find a group of tokens containing semantic information including modifier tokens. The key idea is the cross-attention maps of the modifier tokens are supposed to largely overlap those of biased tokens in ideal conditions for semantic consistency [7, 38]. For example, the cross-attention maps of biased tokens $B_t$ "faces" and "{occupations}" are semantically aligned with the modifier tokens $M_t$ "the", "of", and "{number}". The appropriate threshold can be determined by preserving the linguistic correlation between the semantic regions of these tokens. The ineffective $SR_t$ with no semantic information is initialized with an extremely

large value.

$$L_l = \lambda_e||SR_{B_i}, SR_{B_j}||_2^2 + \lambda_c||SR_{B_i}, SR_{M_p}||_2^2 + \lambda_m||SR_{M_p}, SR_{M_q}||_2^2, \quad (5)$$

where $B_i$ and $B_j$ are different biased noun tokens, and $M_p, M_q$ are different modifier tokens. $\lambda_e, \lambda_c, \lambda_m$ are weights determining the correlation degree of different tokens. The final semantic region $SR_f$ is obtained as follows:

$$SR_f = argmax(||SR_{B_i}, SR_{M_p}||_2^2 + ||SR_{\{B_i, M_p\}}||_2^2). \quad (6)$$

As shown in Fig. 4 (d), the obtained semantic regions are more accurate and stable for different thresholds. Furthermore, the obtained semantic region $SR_f$ presents where the biases are and is further utilized in the Fair Inference stage for debiasing.

### 3.3 Fair Inference

To mitigate generated biases, we aim to represent fair attributes of prompts for each individual in the semantic region $SR_f$ of tokens introducing biases in alignment with the target distribution. For example, for gender debiasing, the fair prompts $c_f$ are "female person" and "male person", which are supposed to be present for individuals with equal probability given the target distribution is equal. Furthermore, the original structure and semantic information of the text prompts $c_p$ are supposed to be preserved. During the inference stage, the latent noise conditioned on $c_p$ is originally predicted by classifier-free guidance [20] as follows:

$$\widetilde{\epsilon}(z_t, c_p) = \epsilon_\theta(z_t) + s_g(\epsilon_\theta(z_t, c_p) - \epsilon_\theta(z_t)). \quad (7)$$

Similarly, to generate the fair attributes $c_f$, the predicted noise is:

$$\widetilde{\epsilon}(z_t, c_f) = \epsilon_\theta(z_t) + s_f(\epsilon_\theta(z_t, c_f) - \epsilon_\theta(z_t)). \quad (8)$$

Directly employ $\widetilde{\epsilon}(z_t, c_f)$ would distort original structural and semantic information of $\widetilde{\epsilon}(z_p, c_p)$. To preserve the original information while presenting fair attributes $c_f$, we consider the semantic dimensions of the noises, which correspond to the upper and lower tail of the noise distribution [4, 43]. These subtle semantic dimensions to original noise preserve the structural and semantic information of $c_f$ while presenting fair attributes of $c_f$. The semantic dimensions of $c_f$ are obtained as follows:

$$\phi(z_t, c_f) = \Omega(\phi, s_f, \omega)(\epsilon_\theta(z_t, c_f) - \epsilon_\theta(z_t)), \quad (9)$$

$$\Omega(\phi, s_f, \omega) = \begin{cases} s_f & |\phi| > \eta_\omega(|\phi|), \\ 0 & otherwise, \end{cases} \quad (10)$$

where $\eta_\omega(|\phi|)$ is the $\omega$-th percentile of $|\phi|$.

Finally, the classifier-free guidance for generating prompt $c_p$ with the semantic dimensions of $c_f$ is obtained as follows:

$$\widetilde{\epsilon}(z_t, c_p, c_f) = \epsilon_\theta(z_t) + s_g(\epsilon_\theta(z_t, c_p) - \epsilon_\theta(z_t)) + \phi(z_t, c_f). \quad (11)$$

For fair generation, we present multiple fair attributes following target distributions $p_i$ for each individual $SR_{f_n}$ from the semantic regions $SR_f$ of tokens introducing biases. The final noise is predicted as follows:

$$\widetilde{\epsilon}(z_t, c_p, c_{f_i}) = \epsilon_\theta(z_t) + s_g(\epsilon_\theta(z_t, c_p) - \epsilon_\theta(z_t)) + \sum_n SR_{f_n} \sum_i^d p_i\phi_a(z_t, c_{f_i}), \quad (12)$$

where $\sum p_i = 1, i \in [1, 2, ..., d]$, $d$ is the number of different fair attributes. $SR_{f_n}$ are the semantic regions obtained by segmentation algorithm [50] for $n$ individuals.

## 4 EXPERIMENTS

In this section, we demonstrate our method outperforms the state-of-the-art debiasing methods to mitigate the social biases with multiple individuals within the image. We conduct both qualitative and quantitative evaluations, and our method achieves superior performance in terms of debiasing as well as preserving the structural and semantic consistency of the original results. Moreover, we apply our method to various scenarios including daily activities, personal descriptors, and style prompts. We further evaluate the debiasing performance with qualitative and quantitative comparisons with the original model.

### 4.1 Experimental Setups

**Social Biases.** We consider social biases including gender, race, and their intersections. For gender, we adopt binary attributes: {Male, Female}, as non-binary attributes are hard determined by outward appearance and are currently challenging for automatic algorithms to identify. For race, we adopt four categories: {Black, White, Asian, Indian} following [46], as these can be better distinguished by classifiers trained on fair racial dataset [21]. We consider the uniform distribution of these attributes and the prompt template is "a photo of the faces of {number} {occupation}" for the multi-face scenarios. "{number}" including cardinal numbers such as "two", "three" and quantifiers "few", "some" and "a group of". We employ the occupations from the International Standard Classification of Occupations.

**Compared Methods.** We compare our method with four baselines. (1) Stable Diffusion with Ethical Interventions [2] (SD-EI), which applies ethical interventions via text prompts to the original model.(2) FairDiffusion [11], which promotes fairness using a fair prompts table and semantic guidance. (3) Unified Concept Editing (UCE) [15], debiasing concepts via multiple edit directions simultaneously by updating the cross-attention layers of the original model. (4) Fine-tune [46], fine-tuning the original model for distribution alignment with a biased gradient. To ensure fair comparisons, all experiments are conducted on Stable Diffusion 1.4 [39].

**Evaluation Metrics.** We utilize the classifiers [46] trained on CelebA [27] and FairFace [21] datasets for gender and race identification. To access social biases in multi-face scenarios, we evaluate the biases of generated results in two ways: across overall results (Bias-W) and within an image (Bias-P).

$$Bias - W = \sqrt{\frac{1}{n_a} \sum_a (freq_a^w - \frac{1}{n_a})^2},$$

$$Bias - P = \sqrt{\frac{1}{n_a} \sum_a (freq_a^p - \frac{1}{n_a})^2}, \quad (13)$$

where $a$ represents different attributes, and $n_a$ denotes the number of attributes. $freq_a^w, freq_a^p$ indicate the frequency of attribute $a$ across all results and within one image. Specifically, the number of $n_a$ is 2 for gender, 4 for race, and 8 for their intersections.

Furthermore, to evaluate the performance of different methods in preserving structural and semantic information of the original results, we employ conventional metrics including SSIM [52], PSNR, and LPIPS [58] to measure the structural consistency, and CLIP [36] and DINO [6] embeddings to assess the semantic consistency via cosine similarity.

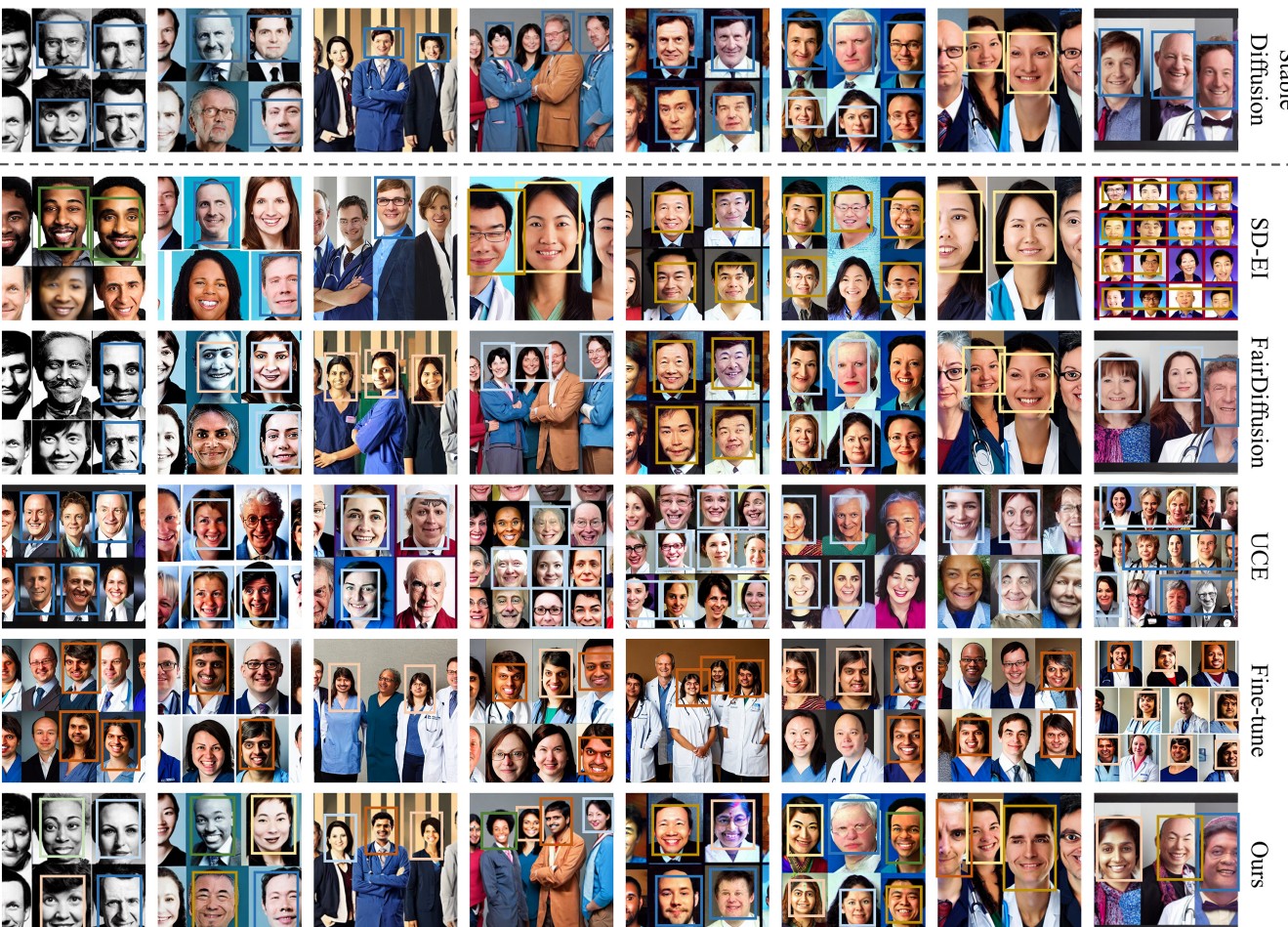

**Figure 5: Qualitative comparison with different methods. Different social groups are highlighted with colored boxes:**"White Female","White Male","Black Female","Black Male","Indian Female","Indian Male","Asian Female","Asian Male".

**Implementation Details.** Our framework is based on the Stable Diffusion 1.4 [39] with default hyperparameters over 50 inference steps. The number of semantic clusters is 4, threshold values $\tau_a \in (0.1, 0.2)$, and the hyperparameters $\lambda_e$=1. During Fair Inference, we set $s_f$=10. All the experiments are conducted on a single V100 GPU.

## 4.2 Qualitative Evaluation

We present the qualitative results of the comparing methods in Fig. 5, and the generated occupation is "doctors". Note that the images in the same column are generated with the same random seeds. For convenience, we highlight the demographic attributes in the figure with colored boxes. More varied colored boxes within an image and across the overall results indicate superior debiasing performance. More qualitative results are provided in the Appendix. As depicted in the figure, Stable Diffusion predominantly generates "White Male" individuals, potentially introducing social biases regarding gender and race. SD-EI exhibits biases towards Asians ("Asian Male" and "Asian Female") and the generated categories are

almost the same within each image. FairDiffusion maintains good structural and semantic consistency. However, it still exhibits identical gender or race within an image, particularly evident in the third, fifth, and sixth columns in Fig. 5. Similar patterns are observed in UCE, which predominantly generates "White Female" instead of equitable representation for other groups. Fine-tune exhibits an inclination toward Indians ("Indian Male" and "Indian Female") and the facial characteristics are similar across individuals, appearing as the same person. As designed for single-face scenarios, current debiasing methods all exhibit homogenized attributes within images containing multiple individuals. Besides, SD-EI, UCE, and Fine-tune preserve less structural and semantic information of the original results. This discrepancy may arise from the disruptions of the original generation process caused by updated components or uncontrolled prompt instructions. In contrast, our method exhibits more diverse and fairer results on different attributes within the images and across all results, as well as the best performance at preserving structural and semantic consistency.

| Method | | Bias-W (↓) | | | Bias-P (↓) | | | Structure Consistency | | | Semantic Consistency (↑) | |
|---|---|---|---|---|---|---|---|---|---|---|---|---|
| | | Gender | Race | G. × R. | Gender | Race | G. × R. | PSNR (↑) | SSIM (↑) | LPIPS (↓) | CLIP-I | DINO |
| | Stable Diffusion | .338 ± .14 | .360 ± .06 | .252 ± .04 | .352 ± .14 | .371 ± .06 | .269 ± .04 | - | - | - | - | - |
| Gender | SD-EI | .188 ± .12 | .337 ± .10 | .214 ± .03 | .242 ± .11 | .356 ± .09 | .242 ± .04 | 13.228 ± 0.54 | .471 ± 0.01 | .453 ± .02 | .823 ± .02 | .808 ± .04 |
| | FairDiffusion | .252 ± .15 | .371 ± .06 | .231 ± .05 | .279 ± .16 | .384 ± .05 | .252 ± .05 | 17.828 ± 1.47 | .692 ± .06 | .248 ± .04 | .897 ± .02 | .917 ± .01 |
| | UCE | .299 ± .22 | .431 ± .01 | .279 ± .05 | .323 ± .21 | .432 ± .00 | .287 ± .05 | 12.377 ± 3.10 | .428 ± .18 | .481 ± .13 | .852 ± .04 | .848 ± 0.07 |
| | Fine-tune | .299 ± .12 | .395 ± .07 | .248 ± .05 | .340 ± .10 | .400 ± .07 | .267 ± .05 | 14.217 ± 0.94 | .515 ± .04 | .412 ± .02 | .859 ± .02 | .842 ± .03 |
| | Ours | .080 ± .04 | .361 ± .07 | .206 ± .02 | .100 ± .03 | .367 ± .07 | .218 ± .03 | 17.929 ± 1.28 | .686 ± 0.05 | .254 ± .03 | .897 ± .01 | .908 ± .01 |
| Race | SD-EI | .260 ± .17 | .322 ± .05 | .212 ± .03 | .300 ± .16 | .362 ± .04 | .242 ± .03 | 12.969 ± 0.44 | .457 ± .03 | .459 ± .02 | .809 ± .02 | .797 ± .05 |
| | FairDiffusion | .310 ± .16 | .371 ± .06 | .243 ± .06 | .340 ± .16 | .294 ± .06 | .242 ± .04 | 16.736 ± 1.74 | .644 ± .06 | .282 ± .06 | .857 ± .02 | .884 ± .02 |
| | UCE | .354 ± .14 | .364 ± .07 | .249 ± .04 | .354 ± .16 | .387 ± .06 | .264 ± .04 | 13.776 ± 1.06 | .509 ± .07 | .418 ± .07 | .839 ± .04 | .833 ± .03 |
| | Ours | .290 ± .05 | .177 ± .01 | .201 ± .02 | .214 ± .03 | .225 ± .02 | .193 ± .02 | 17.190 ± 0.95 | .655 ± .04 | .274 ± .02 | .874 ± .02 | .874 ± .03 |
| G. × R. | SD-EI | .307 ± .21 | .233 ± .05 | .173 ± .04 | .333 ± .19 | .296 ± .05 | .211 ± .03 | 12.299 ± 0.53 | .428 ± .03 | .483 ± .02 | .794 ± .02 | .783 ± .04 |
| | FairDiffusion | .263 ± .14 | .297 ± .10 | .216 ± .06 | .296 ± .13 | .325 ± .08 | .236 ± .05 | 16.087 ± 2.42 | .614 ± .12 | .311 ± .09 | .835 ± .04 | .867 ± .04 |
| | UCE | .325 ± .09 | .403 ± .05 | .271 ± .03 | .354 ± .08 | .408 ± .04 | .286 ± .04 | 11.786 ± 0.78 | .383 ± .08 | .534 ± .08 | .769 ± .07 | .720 ± .12 |
| | Fine-tune | .290 ± .13 | .269 ± .06 | .204 ± .04 | .322 ± .14 | .292 ± .06 | .226 ± .04 | 11.426 ± 0.60 | .376 ± .03 | .519 ± .02 | .793 ± .03 | .808 ± .05 |
| | Ours | .071 ± .02 | .180 ± .01 | .145 ± .02 | .137 ± .10 | .228 ± .02 | .186 ± .03 | 16.698 ± 0.80 | .637 ± .04 | .290 ± .02 | .862 ± .01 | .883 ± .03 |

**Table 1: Quantitative comparisons with different methods. The best results are highlighted in bold and the second to best is highlighted by underline.**

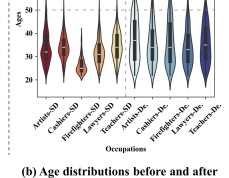

(a) Ablation study of different modules. F.I. and L.A.G. abbreviate Fair Inference and Linguistic-aligned Attention Guidance

(b) Age distributions before and after debiasing for different occupations

**Figure 6: (a) The results of the ablation study. (b) Age distributions of Stable Diffusion and our method for different occupations.**

| | Metric | w/o Both | w/o F.I. | w/o L.A.G. | Full Model |
|---|---|---|---|---|---|
| Bias-W (↓) | Gender | 0.336 | 0.105 | 0.270 | **0.101** |
| | Race | 0.344 | 0.234 | 0.264 | **0.176** |
| | G. × R. | 0.218 | 0.128 | 0.156 | **0.123** |
| Bias-P (↓) | Gender | 0.317 | 0.120 | 0.313 | **0.087** |
| | Race | 0.338 | 0.306 | 0.269 | **0.185** |
| | G. × R. | 0.213 | 0.168 | 0.174 | **0.135** |
| Structure Consistency | PSNR(↑) | - | 11.315 | 11.347 | **12.631** |
| | SSIM(↑) | - | 0.380 | 0.393 | **0.448** |
| | LPIPS(↓) | - | 0.493 | 0.485 | **0.422** |
| Semantic Consistency(↑) | CLIP-I | - | 0.751 | **0.793** | 0.789 |
| | DINO-I | - | 0.795 | 0.833 | **0.847** |

**Table 2: Quantitative ablation results for the components of our model. F.I. and L.A.G. stand for Fair Inference and Linguistic-aligned Attention Guidance respectively.**

## 4.3 Quantitative Evaluation

We generate five hundred images for five occupations using the prompt template involving multiple individuals within the images. We assessed gender, racial, and intersectional biases in terms of overall biases (Bias-W) and biases within images (Bias-P) following Eq. 13. The results are presented in Table 1. As for gender debiasing,

all methods exhibit debiased results in terms of gender, However, UCE and Fine-tune present greater racial bias than the original results, possibly due to fine-tuning the original model with debiasing for one group may inadvertently introduce bias to another. Additionally, they exhibit greater biases within images, indicating a tendency to generate images featuring homogeneous attributes. SD-EI and FairDiffusion demonstrate consistent debiasing performance across three scenarios. However, similar to UCE and Fine-tune, they also focus on the attributes across the entire image instead of individuals, resulting in suboptimal debiasing performance for multiple individuals within the image. In contrast, by identifying the semantic regions from the attention guidance of tokens introducing biases, we effectively locate the biased individuals and debias them while preserving structural and semantic information. Our method achieves consistent lowest biases across all the scenarios, showcasing the best performance at debiasing overall and within images. Furthermore, considering the structural and semantic consistency of the original results, SD-EI, UCE, and Fine-tune exhibit suboptimal quantitative performance. Fair Diffusion better preserves consistency for gender debiasing, while our method outperforms across all scenarios in terms of debiasing, structural integrity, and semantic consistency.

## 4.4 Ablation Study and Analysis

**Ablation Study.** The quantitative and qualitative results of the ablation study are shown in Fig. 6 (a) and Tab. 2. As shown in the results, the generated results without Fair Inference are less similar to the original results in terms of structural and semantic consistency, and without the Linguistic-aligned Attention Guidance, the general debiasing performance declined. The full model performs best at mitigating various biases as well as maintaining structural and semantic consistency.

**Age Distributions.** In addition to biases related to gender, race, and their intersections, we further explore age as another aspect of social biases to facilitate a more comprehensive discussion. However, unlike gender and race, age features continuous discrete values,

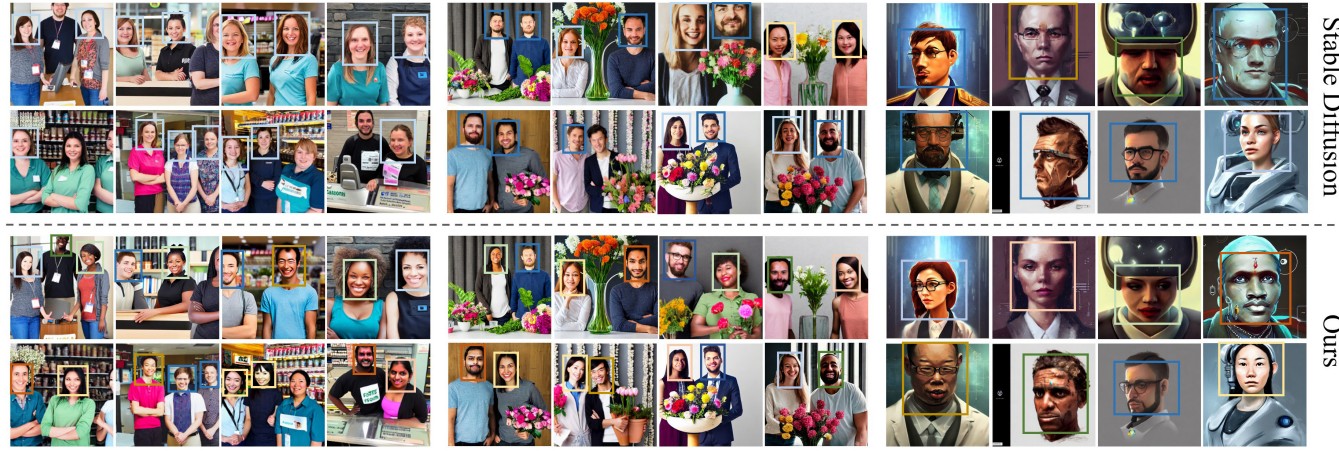

**Figure 7: Qualitative results of Stable Diffusion and our method in other scenarios, including personal descriptors (e.g., smiling), daily activities (e.g., standing next to flowers), and style prompts. The details of prompt templates are reported in the Appendix.**

| Metric | | Descriptors | | Activities | | Styles | |
|---|---|---|---|---|---|---|---|
| | | SD | Debiased | SD | Debiased | SD | Debiased |
| Bias-W (↓) | Gender | 0.271 | 0.173 | 0.473 | 0.149 | 0.314 | 0.138 |
| | Race | 0.228 | 0.208 | 0.417 | 0.257 | 0.311 | 0.131 |
| | G. × R. | 0.165 | 0.143 | 0.320 | 0.220 | 0.192 | 0.077 |
| Bias-P (↓) | Gender | 0.243 | 0.164 | 0.486 | 0.228 | - | - |
| | Race | 0.285 | 0.279 | 0.426 | 0.328 | - | - |
| | G. × R. | 0.227 | 0.220 | 0.326 | 0.265 | - | - |
| Structure Consistency | PSNR(↑) | - | 15.552 | - | 14.602 | - | 17.866 |
| | SSIM(↑) | - | 0.556 | - | 0.572 | - | 0.648 |
| | LPIPS(↓) | - | 0.393 | - | 0.346 | - | 0.328 |
| Semantic Consistency(↑) | CLIP-I | - | 0.902 | - | 0.892 | - | 0.780 |
| | DINO-I | - | 0.918 | - | 0.890 | - | 0.764 |

**Table 3: Quantitative results of Stable Diffusion and ours in other scenarios.**

making it challenging to categorize different groups and lacking a standard criterion. We simply frame age fairness as broader age distributions to assess whether certain groups are underrepresented and ensure diversity in age proportions. To evaluate the age fairness of Stable Diffusion in occupational scenarios, we visualize the age distributions across different occupations in Figure 6 (b). The age of each individual within the images is recognized by a lightweight face analysis model, deepface[1]. As shown in the figure, Stable Diffusion consistently generates individuals under the age of fifty with a focus on ages between thirty and forty, resulting in a limited age distribution and more underrepresented ages. With our debias intervention, the age distributions are broader across different ages and encompass older individuals. Note that our method for age debiasing can also handle each individual within an image.

**Other Scenarios.** To further evaluate the effectiveness of our method, we conduct qualitative (Fig. 7) and quantitative (Tab. 3) experiments on more scenarios with prompt templates involving *personal descriptors* for different occupations: such as "smiling" and "reading", *daily activities*: "persons standing next to flowers", and *style prompts* gathered from LAION-Aesthetics [44] dataset. The

---
[1] https://github.com/serengil/deepface

experimental details and more results are reported in the Appendix. For all these scenarios, our method debiases the original Stable Diffusion considering gender, racial, and intersectional biases. We present more diverse social groups within the images and across all results.

## 5 LIMITATIONS AND ETHICAL CONSIDERATIONS

In this work, the discussion of gender categories is restricted to binary-valued attributes, as automated gender classification currently remains limited to binary-valued gender. Similarly, the categorization of race is simplified to four groups, which requires more discussion when considering other racial categories. Additionally, our method relies on fair prompts for debiasing. If there is a misalignment of the original text-to-image model between prompts and the visual results. Our method may fail to present fair attributes accordingly.

## 6 CONCLUSION

In this work, we introduce a Linguistic-aligned Attention-guided Fair Generation Model to mitigate social biases in terms of gender, race and their intersections. We initially deal with multiple individuals within images of diverse genders and races. Specifically, we propose a Linguistic-aligned Attention Guidance module to identify the semantic regions of tokens introducing biases. Fair Inference is employed to achieve arbitrary target attribute distributions in these regions for each individual, as well as preserving original structural and semantic information. Extensive experimental results and analyses demonstrate our method achieves superior debiasing performance in multi-face scenarios compared with state-of-the-art debiasing approaches. Our method is intended to pave the way for the advancement of more general and robust social-aligned T2I generative AIs.

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
