# OpenReview forum: "Mitigating Social Biases in Text-to-Image Diffusion Models via Linguistic-Aligned Attention Guidance"
_acmmm.org/ACMMM/2024/Conference — MM2024 Poster_

### Official Review · Reviewer_aK1o · 2024-05-24

**Rating:** 5
**Confidence:** 3

**Summary:**

This paper addresses the diversity of social groups in text-to-image diffusion models. Specifically, it aims to identify how biases are generated between text prompts and generated images, locate where the biases are, and mitigate them for each person.

**Strengths:**

+ This proposed method is capable of handling multiple individuals in the same image with diverse genders and races.
+ This paper proposes a Linguistic-aligned Attention Guidance module (via block voting and linguistic alignment) to locate semantic regions/tokens related to biases. Then, Fair inference is employed on the bias regions to generate fair attributes across arbitrary distributions. Original structural and semantic information are preserved.
+ The evaluation results show good visual quality and good isolation from other non-bias semantic tokens.

**Limitations:**

- For cross-attention maps, the use cases of this paper are mainly human subjects and some features are highly overlapped (i.e., human face region). I am particularly interested in how the proposed block voting can improve the cross-attention on concepts with less overlapping in the same region.
- line 906 stated the method relies on fair prompts for debiasing. Please show examples of such cases and how they affect the generated images.

Comments:
Overall, the proposed approach can identify biased regions and perform fair inference to address the bias concerns within an image. I acknowledge that this paper proposed a novel solution with good visual quality to change the sensitive attribute. However, bias is a subjective topic that is highly subjective to the underlying context. I would suggest to position this work as providing tools for the users to smooth the curb related to bias. Sometimes, being fair to all possibilities can be a bias. In this way, users can use these tools to impose the desired distribution of the sensitive bias. In other words, this work provides tools for handling distributional bias from the training dataset.

Others
- [suggestion] For Fig 1, it would be best to use pie to demonstrate the bias for some attributes (especially for binary ones)

**Suitability:**

3

---

### Official Review · Reviewer_EwY7 · 2024-06-03

**Rating:** 3
**Confidence:** 2

**Summary:**

Thia paper introduce a Linguistic-aligned Attention Guidance module consisting of Block Voting and Linguistic Alignment, to effectively locate the semantic regions related to biases.

**Strengths:**

This paper is superior than compared baselines.

**Limitations:**

The novelty is limited, this paper restricted to binary-valued attributes, how about multi-value attribute, the categorization of race is simplified to four groups. The authors only consider diffusion model, how about other generalization model.
I am confused about the motivation,  the motivation seems not direct enough，it looks like the author uses an attention model and mixed some existed methods.
The proposed method lacks theoretical support.
The datasets are small and not enough, the results should compared with recent sota methods.

**Suitability:**

2

---

### Official Review · Reviewer_t4wX · 2024-06-04

**Rating:** 3
**Confidence:** 3

**Summary:**

This work addressed gender and racial bias in text-to-image generation. In this work, bias is considered to be in the diversity of gender and racial attributes of the generated faces. The proposed method locates faces by using cross and self-attention, and then, each face is debiased by using explicit attribute prompts like "female person" or "male person". Results are reported on Stable Diffusion 1.4.

**Strengths:**

- The proposed method can generate more diversity within a group of people in a single image, which is a limitation in current models.
- The proposed method does not require training or fine-tuning.
- The proposed method is compared against multiple debiasing methods and obtains competitive quantitative and qualitative results.

**Limitations:**

**Major**

- The method does not require fine-tuning, but the generation process involves multiple generations for a single image. How much more time and resources does the method need?
- The method only works for simple prompts: "A photo of the faces of {number} {occupations}". In a real scenario with free-form prompts, how can the method detect bias in the self-attention maps?
- There is only a single model used for debiasing, Stable Diffusion 1.4. To show generalization capabilities, the proposed method should be evaluated using several text-to-image generation models. Some models may be difficult to try due to their proprietary nature, but at the very least, different versions of Stable Diffusion, like v2 and sdxl, should be evaluated.

**Minor**

Figures are underexplained:
- In Figure 1, how is frequency computed? For how many images? How are gender and race detected?
- In Figure 2, which method is used in the second row? Please add references
- In Figure 5, why are there some faces without bounding boxes?

Typos
- L269 Priliminary --> Preliminary

**Suitability:**

3

---

### Meta-Review · Area_Chair_LEhw · 2024-07-01

**Recommendation:** Accept (Poster)
**Confidence:** 3

**Metareview:**

The paper proposes a method for debiasing gender and racial biases in diffusion models. The reviewers mention (+) possibility to decrease gender and racial biases which is an important topic, (+) possibility to handle images with multiple humans, (+) competitive results compared to other methods, (+) no need for fine-tuning. The reviewers have some concerns about (-) computational limitations of the method, (-) the number results shown in the paper. Most concerns seem to be cleared up during the rebuttal.

The paper received one positive (weak accept) and two negative (2x borderline reject) initial ratings. After the rebuttal response, the weak accept acknowledged that they think it should be accepted. One of the borderline rejects upgraded their rating to borderline accept, stating that their concerns were solved and that the paper has merit which would make it interesting to discuss at the venue.

The third review is still negative. Unfortunately, the initial review was very low quality, short, and did not include constructive feedback. The authors posted a detailed confidential comment to AC regarding this review, discussing the low quality of the review. I agree that the review quality is rather low, but unfortunately, the reviewer did not revise their review after multiple emails during the initial review phase.
After the rebuttal, it was lowered to "weak reject" with the justification "After reading the rebuttal and the comments from other reviewers, I lowered my rating.", which is a hard to follow judgement considering that the other reviews were accepts... Due to the very low quality of this review and its revisions, I do not believe it should be included in the final judgement.

Following, as both other reviewers argue that this paper has merit for MM, I believe this paper should be accepted for a poster submission, discussing this idea in more detail at the venue.